# Effect of High Efficiency Digestion and Utilization of Organic Iron Made by *Saccharomyces cerevisiae* on Antioxidation and Caecum Microflora in Weaned Piglets

**DOI:** 10.3390/ani13030498

**Published:** 2023-01-31

**Authors:** Yan Zeng, Liwen Jiang, Bingyu Zhou, Yubo Liu, Lingang Wang, Zhijin Hu, Chunping Wang, Zhiru Tang

**Affiliations:** 1Hunan Institute of Microbiology, Changsha 410009, China; 2Laboratory of Bio-Feed and Animal Nutrition, College of Animal Science and Technology, Southwest University, Chongqing 400715, China

**Keywords:** *Saccharomyces cerevisiae*, organic iron, antioxidant capacity, caecum microflora, weaned piglets

## Abstract

**Simple Summary:**

This research assessed the effect of organic iron made by *Saccharomyces cerevisiae* (yeast iron) on digestion, utilization, antioxidation and caecum microflora in weaned piglets. An amount of 20 piglets which had been weaned after 28 days were divided into 4 groups. The basal diet of these 4 groups contained, respectively, 104 mg/kg iron of ferric sulfate or 84 mg/kg, 104 mg/kg or 124 mg/kg iron of yeast iron. The results suggested that a diet containing 84 mg/kg iron of yeast iron has the same effect as a diet containing 104 mg/kg iron of ferric sulfate, and that a diet containing 104 or 124 mg/kg iron of yeast iron is superior to a diet containing 104 mg/kg iron of ferric sulfate.

**Abstract:**

Organic iron is expected to replace inorganic iron used in diets as an iron source. Organic iron possesses high absorption efficiency and low fecal iron excretion. This study aims to study the effect of organic iron produced by *Saccharomyces cerevisiae* (yeast iron) on digestion, utilization, antioxidation and caecum microflora in weaned piglets. In total, 20 piglets that had been weaned after 28 days were divided into 4 groups, each of which followed a different basal diet. The basal diet of each of these 4 groups contained, respectively, 104 mg/kg iron (ferrous sulfate, CON), 84 mg/kg iron (yeast iron, LSC), 104 mg/kg iron (yeast iron, MSC) or 124 mg/kg iron (yeast iron, HSC). This experiment lasted 35 d. The apparent digestibility of iron in LSC, MSC and HMS was higher than that in CON (*p* < 0.01) and the fecal iron content in LSC, MSC and HMS was lower than that in CON (*p* < 0.01). Serum iron contents in LSC, MSC and HMS were higher than that in CON (*p* < 0.01). The iron contents of the heart, lungs, liver, kidney and left gluteus muscle in the MSC and HMS groups were higher than that in CON and LSC (*p* < 0.05). Serum catalase, glutathione peroxidase, superoxide dismutase activity, superoxide anion, glutathione, hydroxyl free radical scavenging rate, total antioxidant capacity, and liver superoxide anion clearance rate and peroxidase in MSC and HMS were higher than that in CON and LSC (*p* < 0.05). The contents of nitric oxide and peroxide of the weaned piglets in MSC and HMS were lower than that in CON and LSC (*p* < 0.05). The abundance of Firmicutes, Blautia and Peptococcus in LSC, HSC and MSC was higher than that in CON (*p* < 0.01). The abundance of Lactobacillus in CON and LSC was higher than that in MSC and HSC (*p* < 0.01). The abundance of Acinetobacter, Streptococcus and Prevotella in LSC, MSC and HSC was lower than that in CON (*p* < 0.01). The results suggested that a diet containing 84 mg/kg iron of yeast iron has the same effect as a diet containing 104 mg/kg iron of ferric sulfate, and that a diet containing 104 or 124 mg/kg iron of yeast iron is superior to a diet containing 104 mg/kg iron of ferric sulfate.

## 1. Introduction

Iron is an important and necessary trace element for the growth development of livestock and poultry. Iron is an important component of molecules undergoing redox reactions in cells. However, iron-poor nutrition, impaired iron absorption from dietary sources (mostly due to interaction with feed components), lowered bioavailability of iron in industrially manufactured feed, or disturbances in the regulation of iron transport, metabolism and resorption lead to the development of iron-deficiency anemia (IDA) in animals or humans [1,2]. Iron deprivation is of limited practical significance in farm livestock. Anemia can develop in young animals, especially piglets, causing illness, subnormal growth and higher mortality [3]. Weaned piglets that grow fast and undergo weaning stress are more prone to iron deprivation in the body and IDA [4].

Iron deprivation can be treated by injection of ferrous salts (e.g., ferrous sulphate), iron dextran (a dextrin) or, more recently, polynuclear ferric hydroxide complexes [5]. The other possibility is supplementation of iron by the oral route using different inorganic ferric or ferrous salts of iron or organic chelates, such as iron proteinate or amino acid iron chelate [6]. However, many inorganic preparations have proven to have poor availability and some have also shown toxic side effects, often attributable to the reduction potential of such compounds and, consecutively, the reduction of ferric iron by superoxide anion. Yeast biomass enriched with iron, when used as a feed supplement, could represent a new and safer solution for prevention of the development of anemia in animals [7]. Iron bound to organic carriers, such as different macromolecules in yeast cells, has been proven to have better absorbability in the organism and is less toxic. In addition, yeast biomass is a good source of proteins, essential amino acids and vitamins [8,9]. Studies have shown that *Saccharomyces cerevisiae* has a unique role in animals’ production performance, digestive ability, immune ability and antioxidation [10], especially in young animals [11]. To date, studies of yeast iron have mainly focused on mice and broilers [12,13], but a few studies have focused on weaned piglets. At present, the main yeast used as feed additive is *Saccharomyces cerevisiae* [11].

Therefore, an organic iron made by *Saccharomyces cerevisiae* (yeast iron) replaces ferrous sulfate in this study. Weaned piglets were used to assess the effect of organic iron made by *Saccharomyces cerevisiae* (yeast iron) on digestion, utilization, antioxidation and caecum microflora. The expectation of this study is that the relative biological potency and the optimum addition amount of yeast iron were evaluated to provide a scientific basis and reference data for the establishment of iron nutritional requirements for weaned piglets, and to provide new ideas for the application of feed additives in animal husbandry. 

## 2. Materials and Methods

### 2.1. Materials

Ferrous sulfate (20.00% iron content, 98% purity) was purchased from Aladdin Shiliu Company; Saccharomyces cerevisiae iron was provided by Hunan Institute of Microbiology (5.16% iron content, 96% purity) and weaned piglets were purchased from Chongqing Zuida Agricultural Development Co., Ltd. (Hechuan, Chongqing, China).

The biochemical assay kits were purchased from Nanjing Jiancheng Biotechnology Engineering Research Institute (Nanjing, China) for the determination of catalase (CAT), glutathione peroxidase (GSH-Px), superoxide dismutase (SOD), xanthine oxidase (XOD), peroxidase (POD), hydroxyl radical scavenging ability, superoxide anion scavenging ability, nitric a dextrin oriron binding capacity (TIBC), ferritin, transferrin, ceruloplasmin and serum iron content. ELISA kits for the determination of glutathione synthase (GSS), heme oxygenase (HO), sulfur redox protein (Trx) in liver and jejunal mucosa was purchased from Quanzhou Ruixin Biotechnology Co., Ltd. (Quanzhou, China).

### 2.2. Experimental Animals, Design, Diets and Feeding Management

An amount of 20 Duroc × Landrace × Yorkshire male piglets (7.01 ± 0.85 kg) weaned after 28 d were obtained from a local commercial swine farm (Chongqing Nongdianshan Agricultural Technology Development Co., Ltd., Chongqing, China). 

All weaned piglets were divided into the following 4 groups according to the different iron supplements in the diets of each group: (1) the basal diet contained 104 mg/kg iron in the form of ferrous sulfate (CON); (2) The basal diet contained 84 mg/kg yeast iron (LSC); (3) The basal diet contains 104 mg/kg yeast iron (MSC); (4) The basal diet contained 124 mg/kg yeast iron (HSC). Each treatment has 5 replicates, and each replicate contained 1 piglet. The trial period was 35 days. All piglets were fed a basal diet formulated according to the NRC (2012) recommendations for various nutrients. The diet ingredients and the nutritional level of diets is shown in Table 1.

On the 21st day of the test, 0.1% titanium dioxide was added to the test diet as an indicator, and the piglets were fed for 10 days. On the 28th day of the test, feces sample were collected for 3 days for the determination of the digestibility of crude protein and iron. After collection, feeding was continued until the end of the test.

Approval for the study was granted by the Ethics Committee of the Experimental Animal Center of Southwest University (IACUC-20210915-07-1). One week before the test, the pig shelter was cleaned and fumigated with formaldehyde. During the trial period, the piglets were kept in individual pens (1.5 m length × 0.5 m width × 0.8 m depth) in a mechanically ventilated and temperature-controlled room (28 ± 1.2 °C) at Animal Experiment Farm, College of Animal Science and Technology, Southwest University. Feed and water were provided on an ad libitum basis.

### 2.3. Sample Collection

In order to more truly reflect the nutrient digestibility of different treatments, we collected feed samples from each treatment diet. The feces of 5 piglets in each group were collected into polythene bags from day 28 to day 31 and stored in a refrigerator at−20 ℃ for digestibility measurement. 

Prior to the morning feed on day 35, a 10-mL blood sample from 5 piglets selected from each group was collected (blood collection vein used was anterior vena cava). The blood samples were undisturbed for 60 min and then centrifuged at 3500 × *g* for 10 min at 4 °C to harvest the serum samples. Serum samples were stored at −20 °C for biochemical analysis and ELISA. 

After blood sampling, 5 piglets from each group were anesthetized with an intravenous injection of sodium pentobarbital (50 mg/kg body weight) and bled by exsanguination. The heart, liver, lung, kidney, adrenal gland, gall bladder, spleen, pancreas, left dorsal longissimus muscle and left posterior gluteal muscle were extracted and weighed. Samples of about 50 g were extracted from the tissue of the heart, liver, spleen, lung, kidney, left dorsal longissimus muscle and left posterior gluteal muscle and collected for the detection of iron content. The contents of the stomach, duodenum (end), jejunum (middle section), ileum (middle section) and cecum (middle section) were collected for the determination of the pH level of the gastrointestinal tract’s contents. The jejunal mucosa was rinsed with cold saline, scraped gently with a scalpel blade and collected. About 5 mL of cecal chyme was collected in a 10 mL sterilized centrifuge tube and was immediately frozen in liquid N_2_, stored at −80 °C for 16S rDNA microflora sequencing.

### 2.4. Determination of Dry Matter and Crude Protein in Feeds and Feces

Dry matter was measured by drying to a constant mass in a forced air oven at 105 °C. Other digesta samples were fractionated by differential centrifugation using the method of Metges et al. [14] and Warren et al. [15]. Total N was determined by the Leco total combustion method, a variation of the Dumas method (AOAC 968.06-2000). 

### 2.5. Determination of Iron Content in Feed, Feces and Tissues

The iron content in the heart, liver, spleen, lung, kidney, left longissimus dorsi muscle and left rear gluteal muscle was determined according to the new technical standard procedure for food hygiene inspection (GB/T 16/265009.13-2003), with use of flame atomic absorption spectrometry (TAS-990 Flame Atomic Absorption Spectrometer). 

### 2.6. Determination of Antioxidant Indexes in Serum, Intestinal Tissue and Liver

Serum CAT, GSH-Px, SOD, XOD, peroxidase, iron hydroxyl radical scavenger capacity, superoxide anion scavenging capacity, glutathione, GSSG, T-AOC, TIBC, ferritin, transferrin and ceruloplasmin concentration were determined according to biochemical assay kits’ instructions. The nitric oxide concentrations, peroxidase activities, hydroxyl radical scavenging capacity, superoxide anion scavenging rate in liver tissue and jejunal mucosa were determined using biochemical assay kits’ instructions. Samples were analyzed by an auto-analyzer (SHIMADZU CL-8000 automatic autoanalyzer Shanghai, China).

### 2.7. Analysis of 16S rDNA Gene Sequencing

The colonic microbial DNA was extracted by the instructions of MOBIO power fecal DNA extraction kit, and colonic microbial 16S rDNA fragments were analyzed by Nianchuan Biological Technology Co. (Hangzhou, China). The V4 region of colonic microbial 16S rDNA was amplified by specific primers 515F (5′-GTGCC AGCMG CCGCG GTAA-3′) and 806R (5′-GGACT ACHVG GGTWT CTAAT-3′) with barcodes. The recovered PCR products were detected and quantified using Qubit 2.0. A 16S rDNA library was constructed by a kit of Illumina TruSeq DNA PCR-Free Prep. Sequencing was performed using a kit from Illumina MiSeq Reagent. PE reads were obtained and spliced using FLASH software. After removing low-quality bases and contaminated joint sequences, data filtering was completed and analyzed using the UPARSE software. The function of colon microbial genes in piglets was predicted and analyzed using PICRUSt software 2.0 from Nianchuan Biological Technology Co. (Hangzhou, China).

### 2.8. Statistical Analysis

The exogenous indicator TiO_2_ was determined to calculate the apparent digestibility of dry matter, crude protein and Fe; the formula was as follows: 1 − B × C/(A × D) where A is the concentration of dry matter, crude protein or Fe in diets (%); B is the concentration of dry matter, crude protein or Fe in diets in fecal samples (%); C is the concentration of TiO_2_ in diet (%); and D is the concentration of TiO_2_ in fecal samples (%). 

All data were presented as means and SEM. All data were subjected to one-way analysis of variance using the general linear model (GLM) procedure of SAS statistical software (SAS Institute, Inc. Cary, NC, USA) according to a completely randomized one-factorial design. SNK test was performed to identify differences among groups. Significance was set at *p* < 0.05.

## 3. Results

### 3.1. Digestibility of Iron and Protein

As shown in Table 2, there were no significant differences in apparent digestibility of crude protein and dry matter among four treatments (*p* > 0.05). The fecal iron content in the LSC, MSC and HMS groups was lower than that in the CON group (*p* < 0.01), whereas there was no difference among LSC, MSC and HMS groups. The apparent digestibility of iron in the LSC, MSC and HMS groups was higher than that in the CON group (*p* < 0.01). The apparent digestibility of iron in the MSC and HMS groups was higher than that in the LSC group (*p* < 0.01), whereas there was no difference between the MSC group and the HSC group (*p* > 0.05).

### 3.2. Serum Iron Index and the Iron Concentration in Organ Tissue

As shown in Table 3, there was no significant difference in serum TIBC, total iron binding capacity, ferritin, transferrin and ceruloplasmin among four treatments (*p* > 0.05). Serum iron content in the LSC, MSC and HMS groups was higher than that in the CON group (*p* < 0.01), whereas there was no difference among the LSC, MSC and HSC group (*p* > 0.05). 

There was no significant difference in the iron contents of the spleen and the left longissimus among four treatments (*p* > 0.05). The iron contents of the heart, liver, kidney and left gluteus muscle in the MSC and HMS groups were higher than that in the CON and LSC groups (*p* < 0.05). There was no difference in iron contents between the CON group and the LSC group or between the MSC group and the HSC group (*p* > 0.05). The iron contents of the lung in the LSC, MSC and HMS groups were higher than that in the CON group (*p* < 0.01), whereas there was no difference between the LSC group, the MSC group and the HSC group (*p* > 0.05).

### 3.3. pH of Intestinal Contents

As shown in Table 4, there was no significant difference in the pH in duodenal contents, jejunal contents, ileal contents, cecal contents and colonic contents among the four groups (*p* > 0.05). The pH value of gastric juice in the LSC, MSC and HMS groups was lower than that in CON (*p* < 0.01), whereas there was no significant difference between the LSC group, the MSC group and the HSC group (*p* < 0.05).

### 3.4. Antioxidant Indexes of Serum, Liver and Jejunum

As shown in Table 5, the serum CAT, GSH-Px, peroxidase and SOD activity, superoxide anion, GSH, superoxide anion removal rate, hydroxyl free radical scavenging rate and T-AOC in the MSC and HMS groups was higher than that in the CON and LSC groups (*p* < 0.05). There was no significant difference between the CON group and the LSC group or between the MSC group and the HSC group (*p* > 0.05). The serum XOD activities in the MSC and HMS groups were higher than that in the CON group (*p* < 0.01), whereas there was no significant difference between the LSC group, the MSC group and the HSC group (*p* < 0.05). The contents of nitric oxide in the MSC and HMS groups were lower than those in the CON and LSC groups (*p* < 0.05), whereas there was no significant difference between the CON group and the LSC group or between the MSC group and the HSC group (*p* < 0.05). There was no difference in serum oxidized glutathione (GSSG) concentration among the four groups (*p* > 0.05).

As shown in Table 6, there was no significant difference in HO, GSS, Trx, NO content, superoxide anion scavenging rate and hydroxyl radical scavenging rate of liver among four treatments (*p* > 0.05). The superoxide anion clearance rate and the POD of liver in the MSC and HMS groups were higher than those in the CON and LSC groups (*p* < 0.05), whereas there was no significant difference between the CON group and the LSC group or between the MSC group and the HSC group (*p* > 0.05). There was no significant difference in NO content, HO, hydroxyl radical scavenging rate and GSS of jejunal mucosa among the four treatments (*p* > 0.05). The peroxidase and superoxide anion clearance rates of jejunal mucosa in the LSC, MSC and HMS groups were higher than that in the CON group (*p* < 0.05), whereas there was no significant difference among the LSC group, the MSC group and the HSC group (*p* > 0.05). The Trx concentrations of jejunal mucosa in the MSC and HMS groups were higher than those in the CON and LSC groups (*p* < 0.01), whereas there was no significant difference between the CON group and the LSC group or between the MSC group and the HSC group (*p* > 0.05).

### 3.5. Cecal Microflora of Weaned Piglets

As shown in Table 7, in general, the Chao1 index, Shannon index, Simpson index, goods_coverage index and Pielou_e index of piglets’ cecal microbes were not significantly affected by the ingestion of yeast iron (*p* > 0.05). The observed OTU indexes of cecal microbes of piglets fed 84 and 124 mg/kg yeast iron were significantly higher than those of the CON group and the MSC group (*p* < 0.05), but there was no significant difference between the CON group, the MSC group, the LSC group and the HSC group (*p* > 0.05).

The PCA map distance of the samples in each group was relatively close, indicating that the microbial composition and structure of the samples were relatively similar (Figure 1A). As shown in Figure 1B, there were 30 species of characteristic flora in the CON group, 34 species in the LSC group, 36 species in the MSC group and 66 species in the HSC group. A Venn diagram was used to explore similarities and differences in microbial communities among groups, showing that the intestinal microbial communities in the caecum contents of piglets in the 4 groups had 205 common OTUs, accounting for 68.3%, 61.9%, 63.7% and 57.3% OTUs in the CON, LSC, MSC, HSC groups, respectively. The results of the Bray–Curtis distance clustering tree structure at the phylum level showed that the difference in intestinal microbial flora, from large to small, is in the following order: LSC, HSC, CON, MSC (Figure 1C).

As shown in Table 7, more than 73% of the bacteria in the cecal contents belonged to Firmicutes and more than 95% of the bacteria in the colonic contents belonged to Firmicutes, Bacteroidetes and Proteobacteria. There was no significant difference in abundance of Actinobacteria, Cyanobacteria and soft-walled phyla among the four treatments (*p* > 0.05). The abundance of Firmicutes in the LSC, HSC and MSC groups was higher than that in the CON group (*p* < 0.01), whereas there was no significant difference among the LSC group, the MSC group and the HSC group (*p* > 0.05). The abundance of Bacteroidetes in the LSC, HSC and MSC groups was lower than that in CON (*p* < 0.01), and the abundance of Bacteroidetes in the MSC and HSC groups were lower than that in the LSC group (*p* < 0.01), whereas there no difference in the abundance of Bacteroidetes between the MSC group and the HSC group. The abundance of Proteobacteria in the HSC and MSC group was higher than that in the LSC and CON groups. The abundance of Proteobacteria in the LSC group was lower than that in the CON group (*p* < 0.01), whereas there was no significant difference between the MSC group and the HSC group (*p* > 0.05). The abundance of Spirochaetota and Tenericutes in the LSC, MSC and HSC groups was lower than that in the CON group (*p* < 0.01), and the abundance of Spirochaetota and Tenericutes in the LSC and HSC groups was lower than that in the MSC group (*p* < 0.01), whereas there was no significant difference between the LSC group and the HSC group (*p* > 0.05).

As shown in Table 7, there was no significant difference in the abundance of Escherichia-Shigella, Staphylococcus, Staphylococcus, Clostridium, Veillonellaceae, Lactococcus, Bifidobacterium, Lachnospiraceae, Roseburia, Ruminococcus Ruminococcus, Faecalibacterium Faecalibacterium and Eubacterium Eubacterium among the four treatments (*p* < 0.05). The abundances of Acinetobacter, Streptococcus and Prevotella in the LSC, MSC and HSC groups were lower than that in the CON group (*p* < 0.01), whereas there was no significant difference in the abundances of Acinetobacter and Streptococcus among the LSC group, the MSC group and the HSC group, and there was no significant difference in the abundance of Prevotella between the LSC group and the MSC group, or between the MSC group and the HSC group (*p* > 0.05). The abundance of Bacteroides in HSC was higher than that in the CON, LSC and MSC groups (*p* < 0.01), whereas there was no significant difference among the LSC group, the MSC group and the CON group (*p* > 0.05). The abundance of Lactobacillus in the MSC and HSC groups was higher than that in the CON and LSC groups (*p* < 0.01), whereas there was no significant difference between the LSC group and the CON group, the MSC group and the HSC group (*p* > 0.05). The abundance of Blautia in the LSC, MSC and HSC group was higher than that in the CON group (*p* < 0.01), and there was no significant difference among the LSC, MSC and HSC groups (*p* > 0.05). The abundance of Peptococcus in the LSC, MSC and HSC groups was higher than that in the CON group (*p* < 0.01). The abundance of Peptococcus in the HSC group was higher than that in the LSC and MSC groups (*p* < 0.01), whereas there was no significant difference between the LSC group and the MSC group (*p* > 0.05).

## 4. Discussion

### 4.1. High Efficiency Digestion and Utilization of Organic Iron Made by Saccharomyces cerevisiae

In the present study, the apparent digestibility of iron in the LSC, MSC and HMS groups was significantly higher than that in the CON group, and the fecal iron content in the LSC, MSC and HMS groups was significantly lower than that in the CON group. These results indicated that yeast iron can be absorbed by the intestinal tract better than inorganic iron and deposited into the tissue; thus, yeast iron is superior to inorganic iron. The reduction of fecal iron content not only reflects the improvement of the iron utilization rate, but also helps to protect the environment. The results of this study were consistent with previous studies; e.g., Männer et al. (2006) found that chelated organic iron and ferric glycinate could reduce iron content in feces more than ferrous sulfate, indicating that organic iron is more bioavailable than inorganic iron [16].

Hemoglobin and serum ferritin are specific indicators reflecting iron storage in the body, and they are also the main indicators for evaluating the iron status of piglets; there is a significant correlation between hemoglobin and serum ferritin, on the one hand, and liver iron content [17]. In this study, the serum iron content in the LSC, MSC and HMS groups was significantly higher than that in the CON group. The iron content of the heart, liver, kidney and left gluteus muscle in the MSC and HMS groups was significantly higher than that in the CON and LSC groups. The iron contents of the lungs in the LSC, MSC and HMS groups were higher than that in the CON group. Serum iron reflects the amount of iron stored in the body and the nutritional status of the body, and it is an effective indicator to detect iron deficiency and iron overload [18]. TIBC refers to the maximum amount of iron that can be bound by transferring in 100mL of serum, also known as serum saturated iron. Ferritin can take up and store iron, while transferrin is directly involved in iron transport and metabolism. Zhang et al. (2016) found that the addition of dipeptide chelated iron to the diet significantly increased the serum iron content, the iron reserves, the serum ferritin and hemoglobin content, and decreased the total iron binding capacity in piglets, indicating that dipeptide chelated iron could improve iron metabolism in piglets to a certain extent [19]. Erikson et al. (1997) fed mice with inorganic iron, yeast iron and iron-deficient diets, respectively, and found that the serum SF content of mice fed iron-containing diets was significantly higher than that of the control group, and the serum SF and STI of the yeast iron groups were significantly higher than those of the inorganic iron group, which indicated that yeast iron could improve the iron storage of piglets more than inorganic iron [20,21]. Overall, the addition of yeast iron was more effective than inorganic iron in increasing the iron content in the serum of weaned piglets, probably because yeast iron was more easily transformed and absorbed in the body than inorganic iron.

Iron is mainly stored in the liver in the form of ferritin in the body. The liver and spleen are the main organs for storing iron. When the body needs it, the storage organs release iron into the plasma. The iron content in the liver is an evaluation of the body’s iron storage. Studies have found that iron content in the liver and kidneys is positively correlated with dietary iron levels [22]. The results of this experiment showed that with the increase of the yeast iron level, the iron content in the heart, liver, lung, kidney and left double gluteus muscle also increased; there were significant differences, which was consistent with the previous experiment results. Compared with the same level of ferrous sulfate, it was found that yeast iron could enhance the ability of internal organs to enrich iron. Winiarska et al. (2016) found that organoiron glycinate can increase the concentration of iron in muscle more efficiently than ferrous sulfate [23]. Consistent with the results of our study, they showed that yeast iron and most organic iron have higher deposition efficiency in tissues than inorganic iron, which is more conducive to the digestion and absorption of animals and has a higher utilization efficiency than inorganic iron.

### 4.2. Effect of Yeast Iron on Antioxidant Function in Weaned Piglets

During the life activities of weaned piglets, redox reactions will occur in the body at any time, and a large amount of reactive oxygen species (ROS) will be generated. An appropriate amount of reactive oxygen species can maintain the normal physiological functions of the body. Proteins, nucleic acids, fats and other biological macromolecules have a strong affinity, thereby destroying the structure of these macromolecules, causing physiological dysfunction of the body, resulting in oxidative stress and, subsequently, a series of diseases. The serum, liver and jejunum contain a variety of antioxidants, such as small molecules, macromolecules and enzymes, with various antioxidant functions, capable of removing ROS and maintaining the dynamic balance of the body’s redox reactions [23,24].

SOD, T-CAT, GSH and GSH-Px are important components of the animal antioxidant system. GSH-Px and SOD are the main indicators for evaluating the oxidative stress of the enzyme system. SOD is an important antioxidant enzyme in the body. Oxidative radicals degrade into hydrogen peroxide (H_2_O_2_), which can cause great harm to the body, while GSH-Px and CAT can decompose H_2_O_2_ into water [25]. GSH is GSH-Px substrate after the action of GSSG; the higher the content of GSH, the lower the content of GSSG [26]. Serum CAT and GSH activities in the MSC group were significantly higher than those in the control group. Serum CAT, GSH-Px and SOD activity, superoxide anion, GSH, superoxide anion removal rate, hydroxyl free radical scavenging rate and T-AOC in the MSC and HMS groups were significantly higher than those in the CON and LSC groups. The superoxide anion clearance rate and the POD of liver in the MSC and HMS groups were significantly higher than those in the CON and LSC groups. The level of T-AOC represents the total antioxidant capacity, which is an important indicator for evaluating nonenzymatic antioxidant capacity [27]. The content of nitric oxide and peroxide in the MSC and HMS groups were significantly lower than those in the CON and LSC groups. Nitric oxide is not only an important cell signaling molecule, but also an immune effector molecule, a vasodilator factor and a gastric mucosal protective factor. Nitric oxide has a complex role in the body, and its balance has an important impact on the health of the body. Nitric oxide can activate NF-κB signaling by the pathway, which causes an inflammatory response and, if elevated nitric oxide is detected, tissue inflammation may occur [28], but an appropriate amount of nitric oxide helps maintains the integrity of the intestinal mucosa [29]. This study indicates that 104 mg/kg *S. cerevisiae* iron reduced inflammation in piglets.

### 4.3. Effect of Yeast Iron on Cecal Microbial in Piglets

The balance of the intestinal microflora of piglets is necessary to maintain a stable intestinal environment [30]. This stability is susceptible to multiple environmental influences; weaning stress may occur, leading to an imbalance in the gut microbial environment, altered microbial flora structure and susceptibility to diarrhea [31,32]. There are significant differences in the diversity and composition of intestinal flora between diarrheal piglets and healthy piglets. Lactobacillus, Bacteroides, Bifidobacterium and other probiotics are dominant in the intestines of healthy piglets. *Escherichia coli*, *Salmonella* and other pathogenic bacteria in the intestines of diarrheal piglets are more likely to become the dominant flora. The intestinal microbes of piglets are mainly concentrated in the cecum and colon, and a relatively stable micro-ecosystem are formed under the interaction of microbes, which are closely related to the health of the body [33]. The relationship and function between the two groups are of great significance. 

The abundance of Bacteroidetes in the LSC, HSC and MSC groups was significantly lower than that in the CON group, and the abundance of Firmicutes in the LSC, HSC and MSC groups was significantly higher than that in the CON group in this study. It has been reported that Firmicutes can produce short-chain fatty acids, improve energy utilization in the diet and regulate immunity, and that the ratio of Firmicutes to Bacteroidetes is generally positively correlated with weight gain [34,35]. Bacteroides are common microorganisms that degrade polysaccharides [36]. This study indicated that Saccharomyces cerevisiae iron is beneficial to cecal microflora in weaned piglets. The abundance of Spirochaetota and Tenericutes in the LSC, MSC and HSC groups was significantly lower than that in the CON group. Most of the spirochetes are pathogenic bacteria or opportunistic pathogens which have the potential to induce an effect on piglet diarrhea [37] and can easily cause diseases, such as spirulina, and dermatitis [38]. The abundances of Acinetobacter, Streptococcus and Prevotella in the LSC, MSC and HSC groups were significantly lower than that in the CON group. An overabundance of Streptococcus can easily cause intestinal flora imbalance. Kassinen et al. (2007) found that there were a large number of streptococci in the feces of patients suffering from irritable bowel syndrome [39]. The abundance of Lactobacillus in the MSC and HSC groups was significantly higher than that in the CON and LSC groups. The abundance of Blautia and Peptococcus in the LSC, MSC and HSC groups was significantly higher than that in the CON group. Benitez-Paez et al. (2020) found that a reduction of Blautella species was related to intestinal inflammation, and its abundance was associated with volatilization. Sex and fatty acid content were negatively correlated, affecting visceral fat accumulation [40].

## 5. Conclusions

In collusion, based on the effects of yeast iron on growth performance, diarrhea rate, antioxidant capacity, anti-inflammatory ability, immunity, intestinal mucosal morphology and colon microflora stability in weaned piglets, these results suggest that a diet containing 84 mg/kg iron in the form of yeast iron has the same effect as one containing 104 mg/kg iron in the form of ferric sulfate, and that a diet containing 104 or124 mg/kg iron in the form of yeast iron is superior to a diet containing 104 mg/kg iron in the form of ferric sulfate. The high efficiency digestion and utilization of yeast iron can replace ferrous sulfate, enhancing antioxidation and improving caecum microflora in weaned piglets.

## Figures and Tables

**Figure 1 animals-13-00498-f001:**
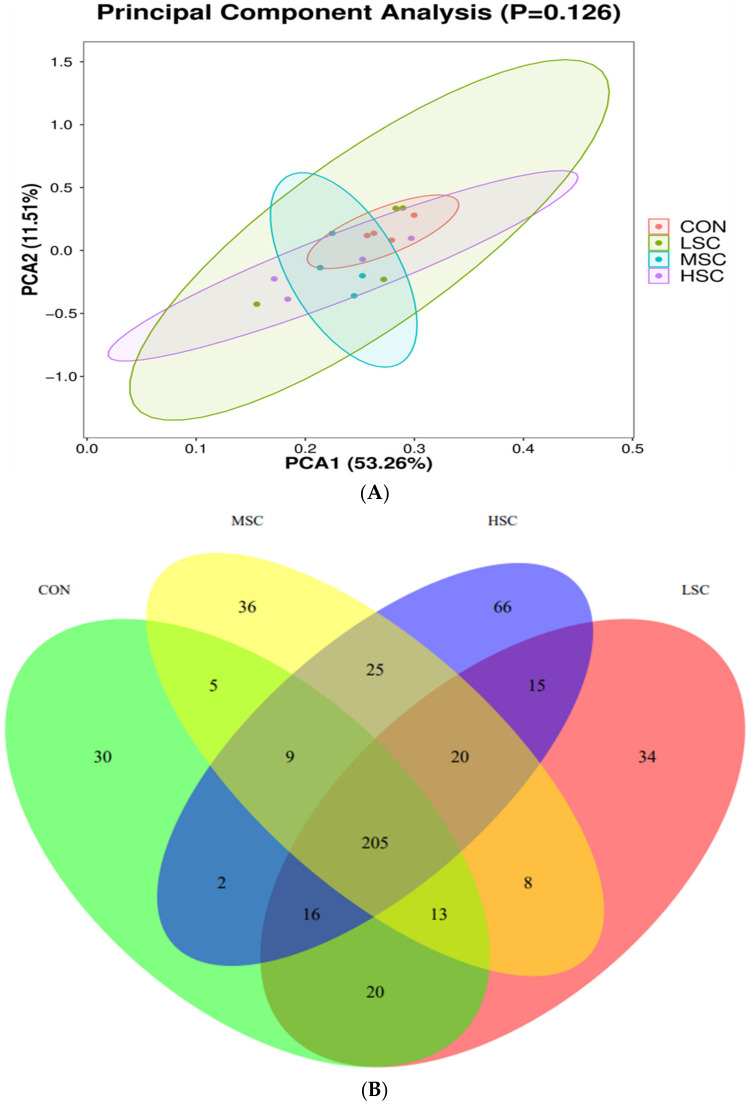
Effects of yeast iron on cecal microbiota communities of weaned piglets. PCA (**A**), Venn diagram (**B**), cluster map (**C**). Note: CON, piglets fed with basal diet; LSC, basal diet contained 84 mg/kg iron made by Saccharomyces cerevisiae; MSC, basal diet contained 104 mg/kg iron made by Saccharomyces cerevisiae; HSC, basal diet contained 124 mg/kg iron made by Saccharomyces cerevisiae.

**Table 1 animals-13-00498-t001:** Diet composition and nutritional level (air-dried basis%).

Ingredients	Content	Nutritional Level	Content
Corn	61.49	Digestive energy (MJ/kg)	14.34
Soybean meal	12.00	Crude protein	18.42
Puffed soybean	6.08	Calcium	0.78
Fish meal	5.00	Total Phosphorus	0.67
Whey powder	10.00	Available Phosphorus	0.40
Fat powder	1.59	Lysine	1.42
Calcium hydrogen phosphate	0.93	Methionine	0.51
Limestone	0.64	Methionine + Cystine	0.78
Salt	0.30	Threonine	0.90
L-Lysine hydrochloride	0.57	Tryptophan	0.21
Methionine	0.21	Crude fiber	2.50
Threonine	0.20	Dry matter	94.75
Premix^1^	1.00		
Total	100.00		

The premix provided the following, per kg, in each diet: 10500 IU VA, 3000 IU VD3, 22.5 IU VE, 3.0 mg VK3, 15 mg pantothenic, 7.5 mg riboflavin, 1.5 mg folic acid, 30.0 mg niacinamide, 3.0 mg thiamine, 4.5 mg VB6, 0.12 mg biotin, 0.03 mg VB12, 100 mg Zn, 4.0 mg Mn, 6.0 mg Cu, 0.3 mg I, 0.3 mg Se, 104 mg/kg Fe in form of ferric sulfate for piglets in the CON group; 84, 104 and 124 mg/kg organic iron made by Saccharomyces cerevisiae for piglets in the LSC, HSC and MSC group, respectively.

**Table 2 animals-13-00498-t002:** Effects of yeast iron on digestibility of iron and protein in weaned piglets.

Items	Treatments ^1^				SEM	*p* Value
	CON	LSC	MSC	HSC		
Fecal iron content (mg/kg)	2816.16 ^a^	2107.45 ^b^	2157.80 ^b^	2153.79 ^b^	78.33	<0.01
The apparent digestibility of iron (%)	73.46 ^c^	80.25 ^b^	86.93 ^a^	87.74 ^a^	1.54	<0.01
The apparent digestibility of crude protein (%)	90.09	89.66	90.15	90.32	0.226	0.797
The apparent digestibility of dry matter	90.15	91.13	91.09	91.52	0.0092	0.188

^1^ Piglets were fed with basal diet (CON), basal diet containing 84 mg/kg iron made by Saccharomyces cerevisiae (LSC), 104 mg/kg iron made by Saccharomyces cerevisiae (MSC) and 124 mg/kg iron made by Saccharomyces cerevisiae (HSC), respectively. ^a,b,c^ Values in the same row with different letter superscripts indicate significant differences (*p* < 0.05).

**Table 3 animals-13-00498-t003:** Effect of yeast iron on serum iron indexes and the iron content in tissues and organs of weaned piglets.

Items	Treatments ^1^				SEM	*p* Value
	CON	LSC	MSC	HSC		
Serum iron content (μmol/L)	84.38 ^b^	109.94 ^a^	110.47 ^a^	111.35 ^a^	3.36	<0.01
Total iron binding capacity (μmol/L)	535.96	546.91	552.83	553.58	3.40	0.239
Ferritin (ng/mL)	66.91	68.71	70.95	71.30	1.08	0.482
Transferrin (g/L)	6.39	6.43	6.33	6.48	0.091	0.953
Ceruloplasmin (U/mL)	16.98	17.31	17.73	17.97	0.248	0.547
The iron content						
Heart (mg/kg)	184.78 ^b^	184.77 ^b^	197.82 ^a^	202.39 ^a^	3.82	<0.01
Liver (mg/kg)	623.24 ^b^	620.48 ^b^	695.89 ^a^	709.83 ^a^	23.23	0.018
Spleen (mg/kg)	749.46	746.47	820.06	832.89	15.28	0.061
Lung (mg/kg)	341.70 ^b^	435.67 ^a^	442.76 ^a^	449.37 ^a^	11.21	<0.01
Kidney (mg/kg)	211.43 ^b^	217.57 ^b^	290.10 ^a^	307.60 ^a^	14.78	0.017
Left gluteus muscle (mg/kg)	48.59 ^b^	46.38 ^b^	55.04 ^a^	57.48 ^a^	1.68	0.039
Left longissimus (mg/kg)	45.79	48.33	54.06	58.31	2.000	0.097

^1^ Piglets were fed with basal diet (CON), basal diet containing 84 mg/kg iron made by Saccharomyces cerevisiae (LSC), 104 mg/kg iron made by Saccharomyces cerevisiae (MSC) and 124 mg/kg iron made by Saccharomyces cerevisiae (HSC), respectively. ^a,b^ Values in the same row with different letter superscripts indicate significant differences (*p* < 0.05).

**Table 4 animals-13-00498-t004:** Effects of yeast iron on the pH value of intestinal contents of weaned piglets.

Items	Treatments ^1^				SEM	*p* Value
	CON	LSC	MSC	HSC		
The pH of gastric juice	4.98 ^a^	2.25 ^b^	2.80 ^b^	1.95 ^b^	0.354	<0.01
The pH of duodenal contents	5.80	5.50	4.80	5.45	0.289	0.705
The pH of jejunal contents	6.20	6.15	6.88	6.20	0.144	0.234
The pH of ileal contents	7.45	7.38	7.28	6.93	0.043	0.107
The pH of cecal contents	6.43	6.35	6.18	6.30	0.049	0.355
The pH of colonic contents	6.85	6.88	6.90	7.05	0.087	0.878

^1^ Piglets were fed with basal diet (CON), basal diet containing 84 mg/kg iron made by Saccharomyces cerevisiae (LSC), 104 mg/kg iron made by Saccharomyces cerevisiae (MSC) and 124 mg/kg iron made by Saccharomyces cerevisiae (HSC), respectively. ^a,b^ Values in the same row with different letter superscripts indicate significant differences (*p* < 0.05).

**Table 5 animals-13-00498-t005:** Effects of yeast iron on serum antioxidant function of weaned piglets.

Items	Treatments ^1^				SEM	*p* Value
	CON	LSC	MSC	HSC		
Catalase (U/mL)	24.14 ^b^	25.70 ^b^	31.96 ^a^	27.84 ^a^	0.997	0.013
Glutathione peroxidase(U/mL)	845.08 ^b^	879.53 ^b^	940.29 ^a^	947.97 ^a^	12.90	<0.01
Superoxide dismutase (U/mL)	338.57 ^b^	343.31 ^b^	453.49 ^a^	465.94 ^a^	17.38	<0.01
Xanthine oxidase (U/mL)	5.89 ^b^	6.58 ^ab^	7.10 ^a^	7.27 ^a^	0.217	<0.01
Peroxidase (U/mL)	12.99 ^b^	12.60 ^b^	19.62 ^a^	18.19 ^a^	0.936	0.021
Superoxide anion removal rate (%)	18.29 ^b^	18.64 ^b^	19.30 ^a^	19.83 ^a^	0.206	0.019
Hydroxyl free radical removal rate (%)	3.84 ^b^	3.97 ^b^	4.43 ^a^	4.54 ^a^	0.092	<0.01
Nitric oxide (μmoL/mL)	0.239 ^a^	0.265 ^a^	0.226 ^b^	0.173 ^b^	0.013	0.044
Glutathione (μmol/mL)	0.216 ^b^	0.225 ^b^	0.256 ^a^	0.272 ^a^	0.016	<0.01
Oxidized glutathione (nmol/mL)	2.71	2.72	1.90	2.10	0.161	0.155
Total antioxidant capacity (μmol Trolox/mL)	0.118 ^b^	0.127 ^b^	0.206 ^a^	0.201 ^a^	0.005	0.015

^1^ Piglets were fed with basal diet (CON), basal diet containing 84 mg/kg iron made by Saccharomyces cerevisiae (LSC), 104 mg/kg iron made by Saccharomyces cerevisiae (MSC) and 124 mg/kg iron made by Saccharomyces cerevisiae (HSC), respectively. ^a,b^ Values in the same row with different letter superscripts indicate significant differences (*p* < 0.05).

**Table 6 animals-13-00498-t006:** Effects of yeast iron on the antioxidant function of liver and jejunal mucosa antioxidant function in weaned piglets.

Items	Treatments ^1^				SEM	*p* Value
	CON	LSC	MSC	HSC		
Liver						
Peroxidase (U/mg protein)	6.62 ^b^	6.69 ^b^	9.86 ^a^	10.56 ^a^	0.537	<0.01
Glutathione synthase (U/g)	0.560	0.565	0.608	0.637	0.017	0.352
Heme oxygenase (U/g)	0.245	0.248	0.251	0.291	0.007	0.055
Sulfur redox protein (ng/g)	1452.08	1486.11	1485.66	1553.43	38.52	0.853
Nitric oxide (μ moL/mg protein)	0.114	0.115	0.104	0.106	0.002	0.150
Superoxide anion removal rate (%)	17.78 ^b^	17.82 ^b^	19.32 ^a^	19.07 ^a^	0.139	0.0135
Hydroxyl free radical scavenging rate (%)	3.89	4.31	4.41	4.02	0.075	0.086
Jejunal mucosa						
Peroxidase (U/mg protein)	4.08 ^b^	8.16 ^a^	8.21 ^a^	8.26 ^a^	0.550	<0.01
Glutathione synthase (U/g)	0.623	0.641	0.634	0.668	0.013	0.696
Heme oxygenase (U/g)	0.325	0.332	0.345	0.371	0.010	0.426
Superoxide anion removal rate (%)	17.85 ^b^	18.53 ^a^	18.85 ^a^	18.90 ^a^	0.141	0.020
Hydroxyl free radical scavenging rate (%)	4.20	4.25	4.64	4.62	0.080	0.074
Nitric oxide (μmoL/mg protein)	0.136	0.130	0.127	0.128	0.007	0.973
Sulfur redox protein (μg/g)	1.07 ^b^	1.16 ^b^	1.42 ^a^	1.49 ^a^	0.049	<0.01

^1^ Piglets were fed with basal diet (CON), basal diet containing 84 mg/kg iron made by Saccharomyces cerevisiae (LSC), 104 mg/kg iron made by Saccharomyces cerevisiae (MSC) and 124 mg/kg iron made by Saccharomyces cerevisiae (HSC), respectively. ^a,b^ Values in the same row with different letter superscripts indicate significant differences (*p* < 0.05).

**Table 7 animals-13-00498-t007:** Effects of yeast iron on alpha diversity index and the relative abundance of cecal microbia in weaned piglets.

Items		Treatments ^1^				SEM	*p* Value
		CON	LSC	MSC	HSC		
Alpha diversity index							
Observed_otus	537.00 ^c^		603.50 ^ab^	562.75 ^bc^	634.50 ^a^	13.20	0.023
Shannon	7.19		6.54	7.08	7.23	0.183	0.561
Simpson	0.967		0.928	0.978	0.972	0.010	0.318
Chao1	563.37		606.13	563.81	636.83	12.19	0.072
Pielou_e	0.782		0.709	0.782	0.778	0.017	0.354
Phylum							
Firmicutes		73.91 ^b^	78.93 ^a^	79.33 ^a^	79.22 ^a^	0.592	<0.01
Bacteroidetes		15.02 ^a^	11.25 ^b^	8.94 ^c^	8.74 ^c^	0.664	<0.01
Proteobacteria		8.54 ^a^	7.80 ^a^	9.89 ^a^	9.97 ^a^	0.260	<0.01
Actinobacteria		1.28	1.33	1.43	1.39	0.036	0.495
Spirochaetota		0.890 ^a^	0.343 ^b^	0.151 ^c^	0.316 ^b^	0.074	<0.01
Tenericutes		0.202 ^a^	0.128 ^b^	0.062 ^c^	0.148 ^b^	0.015	<0.01
**Genus**							
Escherichia-Shigella		1.97	1.45	1.90	1.80	0.112	0.403
Acinetobacter		0.124 ^a^	0.040 ^b^	0.034 ^b^	0.025 ^b^	0.011	<0.01
Streptococcus		7.78 ^a^	4.72 ^b^	4.47 ^b^	4.31 ^b^	0.390	<0.01
Prevotella		11.78 ^a^	7.93 ^c^	8.21 ^bc^	9.43 ^b^	0.433	<0.01
Staphylococcus		0.392	0.366	0.404	0.299	0.028	0.623
Clostridium		10.24	10.61	11.15	9.73	0.203	0.070
Veillonellaceae		4.56	4.25	4.30	4.15	0.095	0.510
Bacteroides		0.408 ^b^	0.316 ^b^	0.361 ^b^	0.650 ^a^	0.039	<0.01
Lactobacillus		3.39 ^b^	3.32 ^b^	4.39 ^a^	4.76 ^a^	0.175	<0.01
Blautia		0.246 ^b^	0.399 ^a^	0.459 ^a^	0.425 ^a^	0.025	<0.01
Peptococcus		0.036 ^c^	0.103 ^b^	0.104 ^b^	0.155 ^a^	0.012	<0.01
Lactococcus		0.566	0.518	0.550	0.547	0.035	0.978
Bifidobacterium		0.141	0.138	0.173	0.210	0.015	0.299
Lachnospiraceae		10.88	12.34	12.04	11.46	0.259	0.198
Roseburia		0.785	0.960	1.048	1.31	0.082	0.140
Ruminococcus		3.71	3.50	3.69	4.23	0.152	0.392
Faecalibacterium		0.497	0.581	0.756	1.17	0.093	0.026
Eubacterium		3.00	3.34	3.53	3.71	0.139	0.318

^1^ Piglets were fed with basal diet (CON), basal diet containing 84 mg/kg iron made by Saccharomyces cerevisiae (LSC), 104 mg/kg iron made by Saccharomyces cerevisiae (MSC) and 124 mg/kg iron made by Saccharomyces cerevisiae (HSC), respectively. ^a,b,c^ Values in the same row with different letter superscripts indicate significant differences (*p* < 0.05).

## Data Availability

The raw data from 16S rDNA sequencing are available at NCBI under the accession number PRJNA827018 accessed on 15 April 2022 at web link (https://www.ncbi.nlm.nih.gov/bioproject/PRJNA827018). The rest of the raw data supporting the conclusions of this article will be made available by the authors without undue reservation.

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
