# Peer review of "Effect of High Efficiency Digestion and Utilization of Organic Iron Made by Saccharomyces cerevisiae on Antioxidation and Caecum Microflora in Weaned Piglets"

_animals, 2023, doi:10.3390/ani13030498_

Round 1
Reviewer 1 Report
Dear Authors,
In general, the manuscript was written very well with proper procedure and comprehensive data. However, several comments are needed to be clarified by the authors as follows:
Major comment
1. The hypothesis should be stated in the introduction.
2. The sex composition of animals should be explained in the text.
3. Is there any consideration for the use of yeast iron level (84 vs. 104 vs. 123 mg/kg) in the present study?
4. Why the author didn’t collect feed intake, final body weight, and gain data? Those data are so important to help the reader to understand real conditions during the feeding trial.
5. In Table 1, is there any information regarding dry matter, organic matter, ether extract, and crude fiber?
6. Why the author didn’t consider the digestibility of DM?
7. Why the author didn’t use polynomial contrast for statistical analysis since the treatment is applied as level?
8. Why the author didn’t prepare negative control (basal diet without any additive/ ferrous sulfate)?
Minor comment
1. The abbreviation of CON, LSC, MSC, and HSC in the ‘simple summary’ should be removed because those abbreviations have no function in the ‘simple summary’.
2. P1 L7, ‘development of IDA’--> IDA should be defined before creating an abbreviation.
3. P1 L23, ‘digestive ability, immune ability, etc’ --> “etc” should be explained in detail, not just 'etc'.
4. ‘Saccharomyces cerevisiae’ should be written in italic.
Author Response
General comments: In general, the manuscript was written very well with proper procedure and comprehensive data. However, several comments are needed to be clarified by the authors as follows.
Authors'responses and locations of the revisions: we sincerely appreciate expert reviewer who have devoted their time to this article and give us many valuable comments and suggestions. We have revised the manuscript according to these suggestion.
Major comment
Comment 1: The hypothesis should be stated in the introduction
Authors'responses and locations of the revisions: we have added the hypothesis in “introduction” (Please the “introduction” in the revised version)
Comment 2: The sex composition of animals should be explained in the text.
Authors'responses and locations of the revisions: Thanks. Owing to your suggestion, The sex composition of animals have been explained in the text. (Please see the revised version).
Comment 3: Is there any consideration for the use of yeast iron level (84 vs. 104 vs. 123 mg/kg) in the present study?
Authors'responses and locations of the revisions: Thanks. We didn’t find any reference, we set the iron level of yeast iron according to the recommendations of NRC (2012).
Comment 4: Why the author didn’t collect feed intake, final body weight, and gain data? Those data are so important to help the reader to understand real conditions during the feeding trial.
Authors'responses and locations of the revisions: Thanks. This study focus on “High Efficiency Digestion and Utilization of Organic Iron”, so we mainly determine indexes about Digestion and Utilization of twenty piglets. Next step we will plan to do another animal experiments to collect feed intake, final body weight, and gain data in piglets farm.
Comment 5: In Table 1, is there any information regarding dry matter, organic matter, ether extract, and crude fiber?
Authors'responses and locations of the revisions: Thanks. We have added DM and crude fiber content in Table 1.
Comment 6: Why the author didn’t consider the digestibility of DM?
Authors'responses and locations of the revisions: Thanks. We have calculated the digestibility of DM.
Comment 7: Why the author didn’t use polynomial contrast for statistical analysis since the treatment is applied as level?
Authors'responses and locations of the revisions: Thanks. Because the control is different from treatments.
Comment 8: Why the author didn’t prepare negative control (basal diet without any additive/ ferrous sulfate)?
Authors'responses and locations of the revisions: Thanks. Because negative control (basal diet without any additive/ ferrous sulfate) rarely occurs in commercial pig farms, 104 ferrous sulfate is commonly used in commercial pig farms. This study focus on replace 104 ferrous sulfate by organic yeast iron.
Minor comment
Comment 1: The abbreviation of CON, LSC, MSC, and HSC in the ‘simple summary’ should be removed because those abbreviations have no function in the ‘simple summary’.
Authors'responses and locations of the revisions: Thanks. We corrected it. (Please see summary in the revised version).
Comment 2: P1 L7, ‘development of IDA’--> IDA should be defined before creating an abbreviation.
Authors'responses and locations of the revisions: Thanks. We defined IDA. (Please see P1 L7 in the revised version).
Comment 3: P1 L23, ‘digestive ability, immune ability, etc’ --> “etc” should be explained in detail, not just 'etc'.
Authors'responses and locations of the revisions: Thanks. We corrected it. (Please see P1 L23 in the revised version).
Comment 4: ‘Saccharomyces cerevisiae’ should be written in italic.
Authors'responses and locations of the revisions: Thanks. We correct it in the whole text (Please see the revised version).
Reviewer 2 Report
Weaned piglets, who grow fast and undergo weaning stress, are more prone to iron deprivation in the body and occur iron deficient anemia (IDA). Yeast biomass enriched with iron, when used as a feed supplement, could represent a new and safer solution for prevention from anaemia development by animals. Iron bound to organic carriers, such as different macromolecules in yeast cells, proved to have better absorbability in the organism and it is less toxic. Studies have shown that Saccharomyces cerevisiae has a unique role in animal production performance, digestive ability, immune ability, etc., especially in young animals. This manuscript made an interesting and complex study on the dietary grapefruit peel powder effect on growth performance, diarrhea, immune function, antioxidant function, ileum morphology and colonic microflora of weaned piglets. These results suggest that the high efficiency digestion and utilization of yeast iron can replace ferrous sulfate to enhance antioxidation and improve caecum microflora in weaned piglets and provide new ideas for the application of feed additives in animal husbandry. Although the topic is important for the animal field, certain aspects require clarifications and corrections, as follows:
1. In the last line of the summary, please replace "and 104 and 124 mg/kg yeas" with "and 104, 124 mg/kg".
2. In the last line of the summary, please replace “iron is superior to 104 mg/kg iron of ferric sulfate.” With “iron is superior to 104 mg/kg iron of ferric sulfate”.
3. Line 82, please replace” or a dextrin or more recently polynuclear ferric hydroxide complexes” with “a dextrin or more recently polynuclear ferric hydroxide complexes”.
4. In line 93, please replace "production performance, digestive ability, immune ability, etc." with "production performance, digestive ability and immune ability, etc."
5. In line 136, the word "The" in "The piglets" should be changed to "the".
6. There is a left bracket after "pentobarbital" in line 156, but no corresponding brackets are visible.
7. Line 252, "and HMS groups was is lower", remove "is ".
8. In line 265, "and in in the MSC", please delete one "in".
9. lines 306,307 should be used in the past tense, replacing "are" with "were"
10. Line 382, "Previous studies" should be in lower case.
11. Line 434, GSH, and GSH-Px should be stripped of the ","
12. Llines 469, 470 and 477, "was" should be changed to "were".
13. Line 45, no ' - ' connection between 28 and d, please remove it.
14. Line 156, 50mg / kg less a ' ) ', please add it.
Author Response
General comments: Weaned piglets, who grow fast and undergo weaning stress, are more prone to iron deprivation in the body and occur iron deficient anemia (IDA). Yeast biomass enriched with iron, when used as a feed supplement, could represent a new and safer solution for prevention from anaemia development by animals. Iron bound to organic carriers, such as different macromolecules in yeast cells, proved to have better absorbability in the organism and it is less toxic. Studies have shown that Saccharomyces cerevisiae has a unique role in animal production performance, digestive ability, immune ability, etc., especially in young animals. This manuscript made an interesting and complex study on the dietary grapefruit peel powder effect on growth performance, diarrhea, immune function, antioxidant function, ileum morphology and colonic microflora of weaned piglets. These results suggest that the high efficiency digestion and utilization of yeast iron can replace ferrous sulfate to enhance antioxidation and improve caecum microflora in weaned piglets and provide new ideas for the application of feed additives in animal husbandry. Although the topic is important for the animal field, certain aspects require clarifications and corrections, as follows: .
Authors'responses and locations of the revisions: we sincerely appreciate expert reviewer who have devoted their time to this article and give us many valuable comments and suggestions. Based on the minor comments of Authors, we have check and correct all wrongs.
Comment 1: In the last line of the summary, please replace "and 104 and 124 mg/kg yeas" with "and 104, 124 mg/kg"
Authors'responses and locations of the revisions: Thanks. We had corrected this sentence. (Please see summary in the revised version).
Comment 2: In the last line of the summary, please replace “iron is superior to 104 mg/kg iron of ferric sulfate.” With “iron is superior to 104 mg/kg iron of ferric sulfate”.
Authors'responses and locations of the revisions: Thanks. We had corrected this sentence. (Please see summary in the revised version).
Comment 3: Line 82, please replace” or a dextrin or more recently polynuclear ferric hydroxide complexes” with “a dextrin or more recently polynuclear ferric hydroxide complexes”.
Authors'responses and locations of the revisions: Thanks. We had corrected. (Please see L82 in the revised version).
Comment 4: In line 93, please replace "production performance, digestive ability, immune ability, etc." with "production performance, digestive ability and immune ability, etc."
Authors'responses and locations of the revisions: Thanks. Owing to your suggestion, we corrected it. (Please see line 93 in the revised version).
Comment 5: In line 136, the word "The" in "The piglets" should be changed to "the".
Authors'responses and locations of the revisions: Thanks. we corrected it. (Please see line 136 in the revised version).
Comment 6: There is a left bracket after "pentobarbital" in line 156, but no corresponding brackets are visible.
Authors'responses and locations of the revisions: Thanks. we corrected it. (Please see line 156 in the revised version).
Comment 7: Line 252, "and HMS groups was is lower", remove "is ".
Authors'responses and locations of the revisions: Thanks. we corrected it. (Please see line 252 in the revised version).
Comment 8: In line 265, "and in in the MSC", please delete one "in".
Authors'responses and locations of the revisions: Thanks. we corrected it. (Please see line 265 in the revised version).
Comment 9: lines 306,307 should be used in the past tense, replacing "are" with "were"
Authors'responses and locations of the revisions: Thanks. we corrected it. (Please see line 306,307 in the revised version).
Comment 10: Line 382, "Previous studies" should be in lower case.
Authors'responses and locations of the revisions: Thanks. we corrected it. (Please see line 382 in the revised version).
Comment 11: Line 434, GSH, and GSH-Px should be stripped of the ","
Authors'responses and locations of the revisions: Thanks. we corrected it. (Please see line 434 in the revised version).
Comment 12: Llines 469, 470 and 477, "was" should be changed to "were".
Authors'responses and locations of the revisions: Thanks. we corrected it. (Please see line 469, 470 and 477 in the revised version).
Comment 13: Line 45, no ' - ' connection between 28 and d, please remove it.
Authors'responses and locations of the revisions: Thanks. we corrected it. (Please see line 45 in the revised version).
Comment 14: Line 156, 50mg / kg less a ' ) ', please add it.
Authors'responses and locations of the revisions: Thanks. we corrected it. (Please see line 156 in the revised version).
Reviewer 3 Report
Introduction
Please add more ref to ref no 10, since at the beginning of that sentence you started with the word studies
add another reference next to ref 12, since ref 12 speaks about rats, you need to add broiler reference also, since you mentioned broilers in the sentence
Material and methods
Sample collection: Please explain and rephrase why feed samples were collected from each treatment diet.
Please also add which vein was the blood collected
Discussion
Please try to also add contradicting research to your current results, because mostly you quoted agreeing results. It will give your discussion depth and make it broad
Author Response
Comment 1: Introduction: Please add more ref to ref no 10, since at the beginning of that sentence you started with the word studies. add another reference next to ref 12, since ref 12 speaks about rats, you need to add broiler reference also, since you mentioned broilers in the sentence.
Authors'responses and locations of the revisions: Thanks. We have added documents related to broilers (Please see line 65 in the revised version).
Comment 2: Material and methods: Sample collection: Please explain and rephrase why feed samples were collected from each treatment diet. Please also add which vein was the blood collected.
Authors'responses and locations of the revisions: Thanks. We have explained why feed samples were collected from each treated diet and supplemented with blood collection vein types. (Please see line 118 and 122 in the revised version).
Comment 3: Discussion: Please try to also add contradicting research to your current results, because mostly you quoted agreeing results. It will give your discussion depth and make it broad.
Authors'responses and locations of the revisions: Thanks. We had revised the Discussion. Because the limitation of letter, this paper didn’t add contradicting research to your current results.